# Discovery of giant unit-cell superstructure in the infinite-layer nickelate PrNiO$_{2+x}$
Jens Oppliger [1] ✉, Julia Küspert [1], Ann-Christin Dippel [2], Martin v. Zimmermann [2], Olof Gutowski[2], Xiaolin Ren[3], Xingjiang Zhou[3], Zhihai Zhu [3], Ruggero Frison[1], Qisi Wang [4], Leonardo Martinelli [1], Izabela Biało[1] & Johan Chang [1] ✉

The discovery of unconventional superconductivity often triggers significant interest in associated electronic and structural symmetry breaking phenomena. For the infinite-layer nickelates, structural allotropes are investigated intensively. Here, using high-energy grazing-incidence x-ray diffraction, we demonstrate how in-situ temperature annealing of the infinite-layer nickelate PrNiO$_{2+x}$ ($x \approx 0$) induces a giant superlattice structure. The annealing effect has a maximum well above room temperature. By covering a large scattering volume, we show a rare period-six in-plane (bi-axial) symmetry and a period-four symmetry in the out-of-plane direction. This giant unit-cell superstructure—likely stemming from ordering of diffusive oxygen—persists over a large temperature range and can be quenched. As such, the stability and controlled annealing process leading to the formation of this superlattice structure provides a pathway for novel nickelate chemistry.

Applications of transition metal oxides span from dental restoration to high-tech semiconductor devices[1]. At the same time, oxide materials host some of the most enigmatic phases of quantum matter. For example, high-temperature superconductivity in the cuprates (copper-oxides) is still an active field of research[2]. A longstanding challenge is to—by design—realize cuprate-physics in other materials[3]. Low-valence nickelates have been a prime candidate for this task. The discovery of superconductivity in doped La$_{1-x}$Sr$_x$NiO$_2$ therefore sparked immediate excitement[4–8]. Much of the following experimental work has been discussed with cuprate physics as reference[9–11]. Experimental studies and calculations agree on a dominant $3d^{9-\delta}$ ground state, but highlighted important differences with respect to cuprates, including a more prominent Mott-Hubbard gap and an active role of rare-earth bands at the Fermi level[9,11–13].

Similarities between nickelates and cuprates were strengthened by the discovery of dispersive magnon excitations, revealing strong antiferromagnetic exchange[14,15]. Another characteristics of cuprates is the presence of two-dimensional charge order in the superconducting planes[16–19]. Therefore, great experimental effort has been put in the search of a similar broken symmetry in nickelates. Recently, evidences of a charge modulation along the Ni-O bonds, was discovered in La-, Nd-, and Pr-based nickelates by resonant x-ray scattering[15,20–23]. However, unlike in cuprates, the order lacks a clear low-temperature dependence[15,21]. Moreover, its dependence on

sample preparation[21] and the unclear role of an epitaxial capping layer[15] question its universality in the family of nickelates. A few proposals to explain the observed modulation include the formation of hydrogen chains[24] or superstructure of re-intercalated oxygen atoms[25]. Most recently, a comprehensive study of NdNiO$_2$ attributed the observed modulation to structural changes induced by patches of oxygen-deficient perovskite phases stemming from partially reduced films[26]. This represents a deviation from the initial interpretation of electronic charge order. Instead, focus is shifted towards the topic of oxygen vacancy ordering and its broader application to perovskite oxygens in general. Oxygen-deficient perovskites, described by the general formula A$_m$B$_m$O$_{3m-x}$, exhibit a wide variety of vacancy patterns and structural configurations that significantly impact their electronic, magnetic, and superconducting properties[27–29]. The ordered removal of oxygen atoms from specific lattice sites leads to the formation of unique structural motifs, such as square pyramids, tetrahedra, and octahedra, depending on the coordination of the metal cations[30]. These vacancy patterns are often systematically arranged in the AO$_{3-x}$ layers, where stacking variations influence the dimensionality and bonding environment.

Here, we present a high-energy, grazing-incidence x-ray diffraction study of PrNiO$_{2+x}$ (PNO) with crystalline and amorphous SrTiO$_3$ (STO) capping layer. In contrast to resonant diffraction, this technique covers a large scattering volume across many Brillouin zones. Our main finding is a

[1]Physik-Institut, Universität Zürich, Zürich, Switzerland. [2]Deutsches Elektronen-Synchrotron DESY, Hamburg, Germany. [3]Beijing National Laboratory for Condensed Matter Physics, Institute of Physics, Chinese Academy of Sciences, Beijing, China. [4]Department of Physics, The Chinese University of Hong Kong, Shatin, Hong Kong, China. ✉e-mail: jens.oppliger@physik.uzh.ch; johan.chang@physik.uzh.ch

stable, giant unit cell emerging upon in-situ thermal heating above ambient temperature. In the $NiO_2$ plane, a rare period-six translational symmetry occurs with a period-four stacking order in the out-of-plane direction. This giant unit-cell superstructure remains stable over a large temperature range and emerges irrespectively of crystalline or amorphous capping. As such, it represents a fundamentally novel structure—most likely originating from ordering of diffusive oxygen. Quenching this structure to low temperatures promises access to new nickelate chemistry.

## Results

Three distinct PNO thin films on an STO substrate, each with either a crystalline (C) or amorphous (A) STO capping layer, were measured. Dimensions and room temperature lattice parameters of the thin films and the substrate are provided in Table 1. Our grazing-incidence diffraction geometry is schematically illustrated in the lower part of Fig. 1a. Rotating the sample around the direction perpendicular to the scattering plane ($\omega$) allows collection of a large three-dimensional scattering volume, covering dozens of Brillouin zones, as demonstrated by data recorded on a PNO thin film grown on an STO substrate with crystalline STO capping layer (sample C1), shown in the upper part of Fig. 1a. In Fig. 1b–d, we display two-dimensional slices of the scattering volume. From such slices, fundamental Bragg reflections yield information about lattice parameters and translational symmetry breaking. As common for epitaxially strained growth, the in-plane lattice parameters of thin film and substrate are identical within our experimental resolution. By contrast, along the out-of-plane $c$-axis direction, lattice parameters of substrate and PNO film are clearly different—see Fig. 1c, d. Throughout the manuscript, reciprocal lattice units (and super-lattice reflections) are based on the PNO film system.

We define fundamental thin film Bragg peak positions as $\tau = (h_B, k_B, \ell_B)$ – with $h_B$, $k_B$, and $\ell_B$ being integers. Upon heating above room temperature, we discover the emergence of additional commensurate reflections. These reflections occur at $Q_o = \tau(q_o^{a,b} + q_o^c)$ with

$$q_o^{a,b} = (\delta_h, \delta_k, 0) \quad \text{and} \quad q_o^c = (0, 0, \delta_\ell) \qquad (1)$$

and in-plane commensurabilities $\delta_h \approx \delta_k \approx 1/6$ and out-of-plane commensurability $\delta_\ell \approx 1/4$. Examples of superlattice peaks at $Q_o = (\pm 1/6, 1/2, 7/4)$ and $Q_o = (1/2, \pm 1/6, 7/4)$ are highlighted by circles and squares in Fig. 1b. Notice that the $Q_o = (1/6, 1/6, \ell_B \pm \delta_\ell)$ reflections are either weak or symmetry forbidden.

In Fig. 2, we focus on the $(h, 1/2, \ell)$ and $(h, 1, \ell)$ scattering planes respectively for sample C1 – see Supplementary Fig. 1 for equivalent $(1/2, k, \ell)$ and $(1, k, \ell)$ data. Reflections in both scattering planes display the same temperature dependence. Initially, the superlattice peaks emerge and are enhanced upon heating above room temperature. Note that at room temperature, the out-of-plane scan reveals a broad peak centred around $\delta_\ell \approx 1/3$—see Fig. 2f. This is also in agreement with previous resonant x-ray scattering studies[21,22,26]. Upon heating, the out-of-plane commensuration changes to a sharp peak with $\delta_\ell \approx 1/4$ as shown in Fig. 2c, f. This phase with quarter commensuration furthermore displays a much longer out-of-plane correlation length $\xi_c$, indicating an improved stacking order.

In Fig. 3, we summarize the temperature dependence of the superlattice peaks for samples C1 and A1, namely systems with crystalline and amorphous STO capping layer, respectively. Figure 3a, b shows that the PNO in-

### Table 1 | Studied film systems

| Label | Film system | d [nm] | a [Å] | b [Å] | c [Å] |
|---|---|---|---|---|---|
| C1 | Cryst. STO capping | 4.0 | \| | \| | ∅ |
| | $PrNiO_{2+x(C1)}$ | 7.0 | 3.885 | 3.905 | 3.450 |
| | STO substrate | ∞ | \| | \| | 3.895 |
| C2 | Cryst. STO capping | 4.0 | \| | \| | ∅ |
| | $PrNiO_{2+x(C2)}$ | 7.1 | 3.883 | 3.905 | 3.290 |
| | STO substrate | ∞ | \| | \| | 3.895 |
| A1 | Amorph. STO capping | 5.0 | \| | \| | ∅ |
| | $PrNiO_{2+x(A1)}$ | 7.6 | 3.900 | 3.915 | 3.305 |
| | STO substrate | ∞ | \| | \| | 3.900 |

Thicknesses (d) and room temperature lattice parameters of the three capped film systems used for this study. In-plane parameters of the substrate, film, and capping are identical within the experimental sensitivity. Given that the capping layer is very thin, it is not possible to identify the $c$-axis lattice parameter and hence this entry is indicated by ∅. Films C2 and A1 yield a $c$-axis lattice parameter comparable to cap-free $PrNiO_2$ films[6,37]. Based on the $c$-axis lattice parameter, we deem that $x(C2) \sim x(A1) < x(C1) \approx 0$.

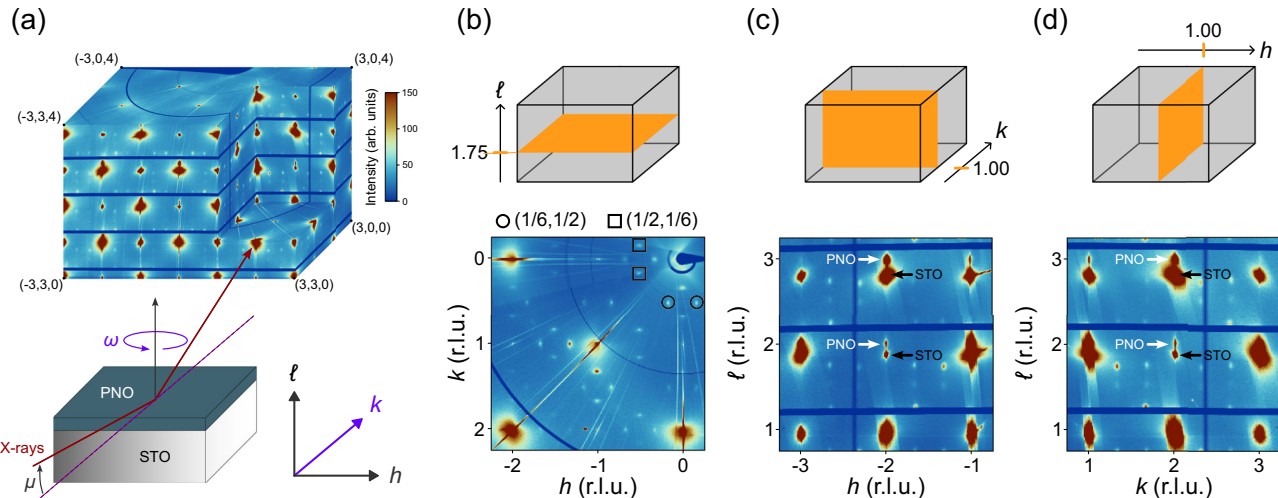

**Fig. 1 | High-energy grazing-incidence x-ray diffraction on a $PrNiO_{2+x}$ thin film.** **a** Schematic illustration of the diffraction geometry. A three-dimensional scattering volume is recorded by high-energy grazing-incidence (angle $\mu$) x-rays, diffracted on a horizontal film, which is rotated around its vertical axis ($\omega$). **b–d** Two-dimensional cuts displayed schematically (top) together with intensity maps of respectively the $(h, k, 1.75)$, $(h, 1, \ell)$, and $(1, k, \ell)$ scattering planes measured at 386 K in terms of reciprocal lattice units (r.l.u.) (bottom). Diffracted intensities are visualized using a linear false color scale. Fundamental Bragg reflections of the $PrNiO_{2+x}$ (PNO) film and the $SrTiO_3$ (STO) substrate are indicated by arrows in (**c**, **d**). Principle superlattice reflections are highlighted with square and circular symbols in (**b**).

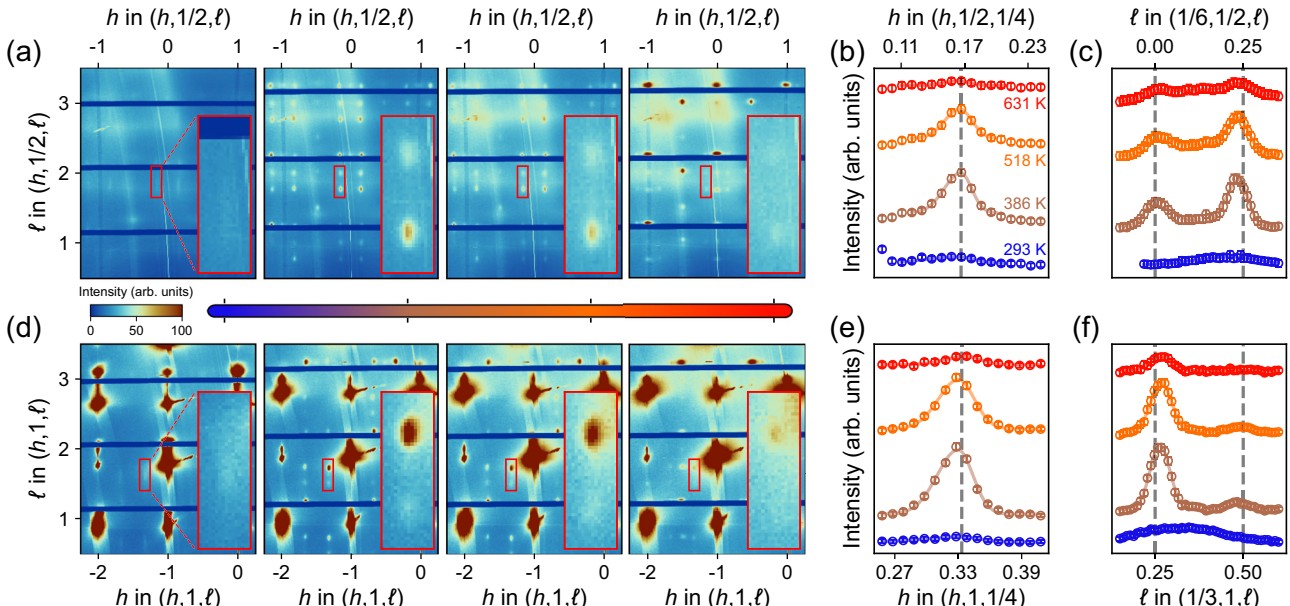

**Fig. 2 | Thermally-induced superlattice structure in a PrNiO$_{2+x}$ thin film.** **a, d** Diffraction intensities (linear false color scale) in the $(h, 1/2, \ell)$ and $(h, 1, \ell)$ scattering planes as a function of temperature. The four temperatures are indicated in panel (**b**). The most intense peaks stem from fundamental Bragg peaks of the SrTiO$_3$ substrate and the PrNiO$_{2+x}$ thin film. Selected superlattice peaks are highlighted by the red rectangular boxes. **b, c** One-dimensional $h$ (in-plane) and $\ell$ (out-of-plane) scans through the superlattice reflections in (**a**) for temperatures as indicated. **e, f** Equivalent $h$ and $\ell$ scans but through the superlattice reflections in (**d**). Solid lines are Gaussian profiled fits with a sloping background. Error bars reflect counting statistics.

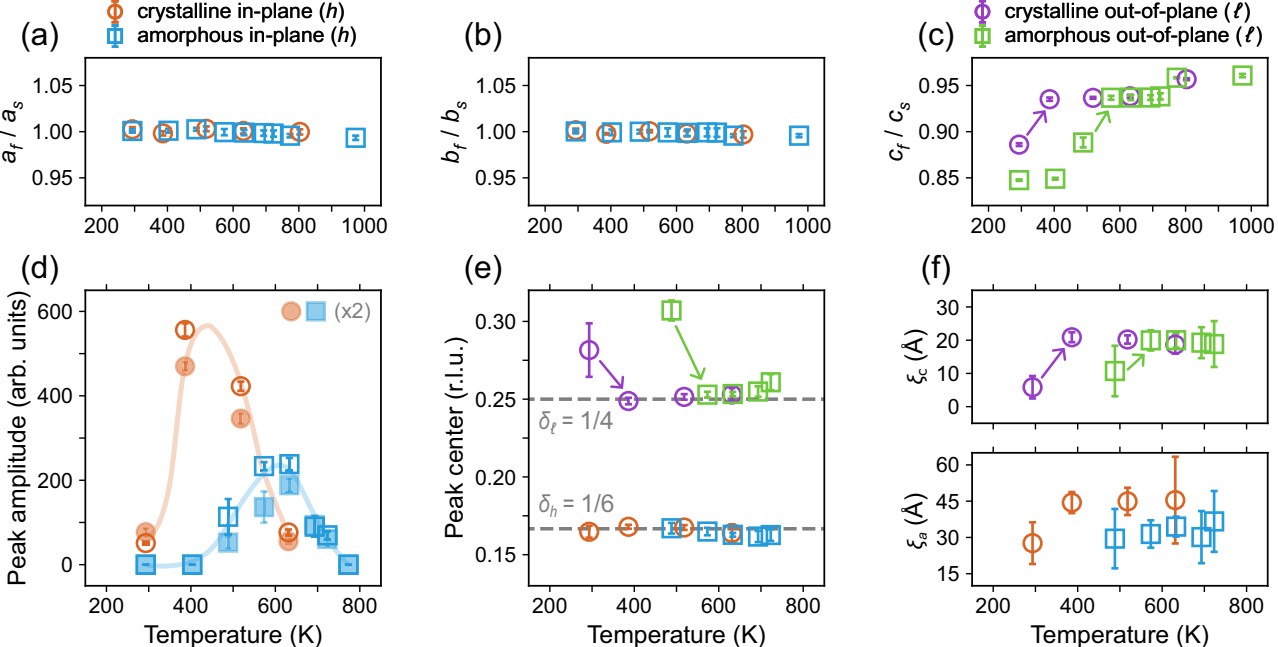

**Fig. 3 | Temperature and capping layer dependence of the superlattice structure.** **a–c** PrNiO$_{2+x}$ thin film lattice constants $z_f$—normalized to the (approximately constant) SrTiO$_3$ substrate lattice constants $z_s$, with $z = a, b, c$ for films with crystalline and amorphous capping layers. **d** Peak amplitude versus temperature of selected reflections $Q_o = (1/6, 1/2, 7/4)$ (filled markers) and $Q_o = (1/3, 1, 7/4)$ (empty markers) for films with crystalline and amorphous SrTiO$_3$ capping. Amplitude of $Q_o = (1/6, 1/2, 7/4)$ has been scaled by a factor of two. Solid lines are guides to the eye. **e** In-plane $\delta_h$ and out-of-plane $\delta_\ell$ commensuration plotted versus temperature. (**f**) In-plane $\xi_a$ and out-of-plane $\xi_c$ correlation lengths versus temperature. Data in (**e**) and (**f**) is averaged over the two reflections separately for crystalline and amorphous capping. Error bars represent one standard deviation obtained from a least squares fitting procedure.

plane $(a, b)$ lattice constants are essentially temperature independent for both samples. By contrast, the out-of-plane $c$-axis lattice constant shows a step-like temperature dependence—see Fig. 3c. Roughly at this step (highlighted with arrows), the quartet peaks at $Q_o$ appear. We note that due to the correlation between the sudden change in the $c$-axis lattice constant and the

emergence of superlattice peaks, the corresponding chemical structure is likely different from stoichiometric PrNiO$_2$.

Both samples C1 and A1 show the emergence of the superlattice peak. For sample C1, the superlattice peak is already present at room temperature. For sample A1, the onset of the superlattice peak is shifted towards higher

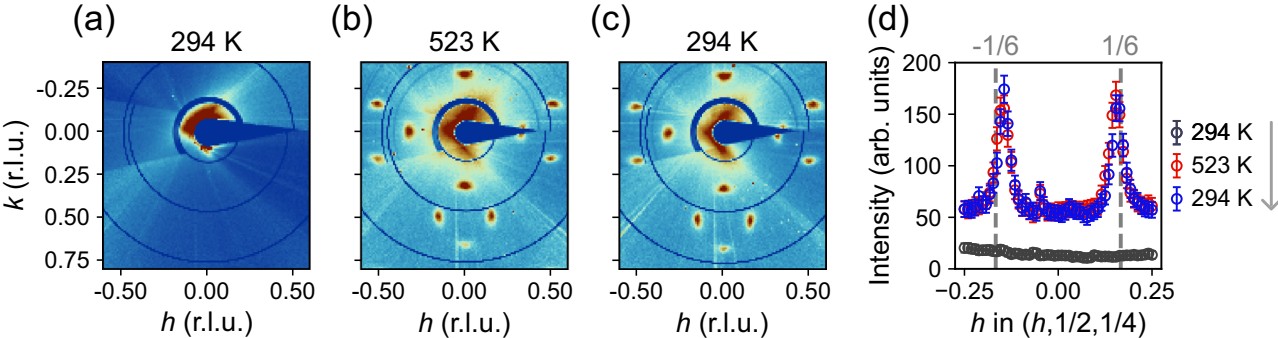

**Fig. 4 | Temperature quenching of the superlattice structure. a–c** Diffracted intensities within the ($h$, $k$, 1.75) scattering plane in terms of reciprocal lattice units (r.l.u.) for temperatures as indicated. **d** Corresponding $h$-scans through ($h$, 1/2, 1/4), demonstrating how the annealing-induced symmetry breaking can be quenched. The quenching experiment was carried out on a PrNiO₂ thin film that was kept under vacuum conditions before being introduced into the controlled helium atmosphere in the XRD chamber. The sample was measured in a full cycle of heating and cooling back to room temperature. Error bars reflect counting statistics.

temperatures where it first emerges with $\delta_h \approx \delta_k \approx 1/6$ and a broad peak with $\delta_\ell \approx 1/3$. At even higher temperatures, the out-of-plane commensuration—like in the case of crystalline capping—manifests as a sharp peak with $\delta_\ell \approx 1/4$. For temperatures above 600 K the quartet peaks are suppressed and eventually vanish at temperatures above 800 K as shown in Fig. 3d. High enough temperatures therefore seem to reverse the topotactic reaction and return the film system to the PrNiO₃ cubic perovskite structure. This is confirmed by laboratory $2\theta$ scans shown in Supplementary Fig. 2. For the sample with amorphous capping, the onset temperature of the quartet peaks is shifted by around 100 K and is less pronounced compared to the sample with crystalline capping. Interestingly, after entering the ordered state, the in-plane $\delta_h \approx \delta_k \approx 1/6$ and out-of-plane $\delta_\ell \approx 1/4$ commensurations show little to no temperature dependence – see Fig. 3(e). The same holds for both in-plane and out-of-plane correlation lengths—see Fig. 3f.

In Fig. 4, we show how the annealing-induced superlattice can be quenched using a PNO thin film with crystalline STO capping layer (sample C2). Initially, at room temperature, no translational symmetry breaking is observed. Upon heating—in this case to 523 K—superlattice reflections emerge. Once returning to room temperature, the superlattice structure remains. It is therefore possible to quench the superlattice structure to lower temperatures.

## Discussion

Our results suggest that between the known ANiO₂ and ANiO₃ crystal structures, there exist—at least two—superlattice structures with gigantic unit cells. Previous resonant x-ray scattering studies have reported an in-plane modulation with a periodicity of three lattice units and poor inter-plane correlation[15,20–23]. The initial charge ordering interpretation has recently been contested by experimental evidence pointing to oxygen ordering as the source of the modulation. Specifically, it has been suggested that residual apical oxygen (ANiO₂₊ₓ) may distribute with a quasi two-dimensional period three modulation[26].

In this work, we show that PrNiO₂₊ₓ ($x \approx 0$) can be annealed to form a three-dimensional ordering with an in-plane period six and an out-of-plane period four. As such, a single apical oxygen atom per $6 \times 6 \times 4$ (original) unit cells generates the observed symmetry breaking. The in-plane period six is consistent with previous x-ray observations, in which the superlattice structure was interpreted as having a period three. The here reported three-dimensional superlattice is composed of two independent orderings: a fundamental two-dimensional ordering and different stacking patterns. This is reminiscent of two-dimensional charge orderings in the cuprates or dichalcogenides where different stacking orders frequently occur[31,32]. In our particular case, we report a fundamental in-plane order that stacks with a (short-range) period three or a (long-range) period four along the $c$-axis. Based on our diffraction experiment, it is not possible to distinguish checkerboard (biaxial) from twinned stripe order.

Irrespective of exact symmetry breaking, the reflections contain valuable information about the nature of the ordering. The observed superlattice reflections are intense – only one or two orders of magnitude weaker than the fundamental Bragg peaks of the thin film (see Fig. 1). This suggests that the symmetry breaking stems from a strong ordering tendency[33]. This would be atypical for charge density waves that typically manifest by weak reflections. Yet, the quenching effect suggests that the observed symmetry breaking goes beyond a standard crystal structure phase transition.

It is possible that oxygen diffuses from the substrate and/or capping layer to the film or that the topotactic process left a residual (homogeneous or heterogeneous[23,26]) apical oxygen occupation. In both cases, high temperatures will enhance oxygen diffusion and promote an oxygen annealing process as seen for example in YBa₂Cu₃O₆₊ₓ[31] and related oxides[30]. The first scenario seems more plausible, considering the absence of room-temperature super-lattice peaks in the thin films of samples A1 (Fig. 3d) and C2 (Fig. 4). On the contrary, the presence of room-temperature superlattice peaks in the thin film of sample C1 (Figs. 2 and 3d) can be explained by oxygen diffusion prior to the x-ray experiment (see Methods section). Oxygen diffusion would render our PrNiO₂ film off-integer stoichiometric by occupying vacant apical oxygen positions. Such a partial apical oxygen occupation is consistent with the observed $c$-axis extension—see Fig. 3c.

We stress that due to the weak form factor, apical (or in-plane) oxygen alone can not explain the observed structure factor. However, apical oxygen inclusions may induce Ni and Pr distortion patterns. Due to the large atomic mass, Pr distortions are likely to dominate the structure factor. A structural refinement would be an interesting future extension of this work.

An open pressing question is as to why the giant $6 \times 6 \times 4$ unit cell manifests over a 300 K temperature range, irrespective of crystalline or amorphous capping. In principle, oxygen diffusion would produce an arbitrary oxygen stochiometry. Our observation of a stable giant super-structure implies a significant down-scaling of the Brillouin zone. As such, it is possible that the superstructure induces an electronic state with favorable energetics. This hypothesis therefore implies the existence of two fundamentally different ground states of PrNiO₂₊ₓ. The fact that spin excitations —in ANiO₂—are not observed in combination with this symmetry breaking[15], supports this rationale. It would thus be of great interest to quench the giant superlattice structure to low temperature for studies of its electronic structure and properties.

## Methods
### Film systems
Three samples of capped films of PrNiO₂₊ₓ have been studied, namely two samples with crystalline capping (labeled C1 and C2) and one sample with amorphous capping (labeled A1). Thicknesses of films and cappings are indicated in Table 1 along with lattice parameters. The thin films were grown on a (001)-oriented SrTiO₃ substrate by pulsed laser deposition.

During growth, the substrate temperature was kept at 600 °C under an oxygen partial pressure of 150 mTorr. After topotactic reduction (390 °C for 2 h), the pervoskite phase is transformed into an infinite-layer phase. The samples (with dimensions $5 \times 5 \times 0.5$ mm$^3$) were cleaned with isopropyl and afterwards dried with compressed air before transferring them into the experimental chamber. Prior to the measurement, one thin film (C1) with crystalline capping was exposed to air for over a day. The other thin films (C2 and A1) with crystalline and amorphous capping were measured immediately after they have been removed from an inert atmosphere.

## Diffraction experiments

High energy x-ray diffraction experiments were carried out at the second experimental hutch (EH2) of the P07 beamline[34–36] at the PETRA III storage ring (DESY, Hamburg). 73 keV x-rays with grazing-incidence geometry ($\mu = 0.05°$) and a Detectris Pilatus3 X CdTe 2M detector were used. For each scan, an angular range of 200° ($\omega$ in Fig. 1a) has been covered using a total of 2000 frames. Each frame therefore corresponds to an angular range of 0.1°. The exposure time per frame was set to 0.05 s. The samples were kept in a helium atmosphere with constant flow rate. Temperature was controlled by a resistive heating plate. The temperature heating (cooling) ramp rate was ~ 2.5 (1.5) K min$^{-1}$. After reaching the target temperature, the samples were thermalized to thermodynamic equilibrium before starting the measurement.

## Data analysis

Detector images are reconstructed into reciprocal space and shown two-dimensional data slices are integrated over 0.1 reciprocal lattice units along the slicing direction. The peaks of the one-dimensional line profiles for the sample with (crystalline) amorphous capping layer are fitted with a (linear) quadratic background and a (split) Gaussian function. Correlation lengths when using a split Gaussian function are obtained from the average of the standard deviations of the Gaussians.

## Data availability

All experimental data are available upon reasonable request to the corresponding authors.

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

**Article**

## Acknowledgements

J.O., J.K., L.M., and J.C. acknowledge support from the Swiss National Science Foundation (200021_188564). J.O. acknowledges support from a Candoc grant of the University of Zurich (Grant no. K-72334-06-01). J.K. is further supported by the PhD fellowship from the German Academic Scholarship Foundation. I.B. and L.M. acknowledges support from the Swiss Government Excellence Scholarship. Z.Z. acknowledges the support from the National Natural Science Foundation of China (Grant No. 12074411), the National Key Research and Development Program of China (Grant Nos. 2022YFA1403900 and 2021YFA1401800). Q.W. is supported by the Research Grants Council of Hong Kong (ECS No. 24306223), and the CUHK Direct Grant (No. 4053613). Parts of this research were carried out at beamline P07 at DESY, a member of the Helmholtz Association (HGF). The research leading to this result has been supported by the project CALIPSOplus under the Grant Agreement 730872 from the EU Framework Programme for Research and Innovation HORIZON 2020.

## Author contributions

X.R., X.J.Z., and Z.Z. grew the $PrNiO_2$ films. Sample preparation for the x-ray experiments were organized by I.B. and J.K. J.O., J.K., O.G., A.C.D., M.v.Z., L.M., and J.C. carried out the experiment. J.O. carried out the data analysis with assistance from M.v.Z., J.K., I.B., L.M., R.F., and J.C. The project was conceived by Q.W. and the manuscript was written by J.O. and J.C. with assistance from all authors. J.O. and J.K. contributed equally.

## Competing interests

The authors declare no competing interests.
