## [Transparent Peer Review file · Communications Materials]

Discovery of Giant Unit-Cell Super-Structure in the Infinite-Layer Nickelate PrNiO_{2+x}

Corresponding Author: Mr Jens Oppliger

This manuscript has been previously reviewed at another Nature Portfolio journal. This document only contains reviewer comments and rebuttal letters for versions considered at Communications Materials.

Version 0:

Decision Letter:

Dear Mr Oppliger,

Thank you for transferring your revised manuscript, "Discovery of Giant Unit-Cell Super-Structure in the Infinite-Layer Nickelate PrNiO_2 ", to Communications Materials. It has now been seen by the original 3 referees, whose comments are appended below.

As you will see, they find your work greatly improved, but a few important points are still raised. In particular, the discussion about oxygen-ordering vs charge-ordering should be more fairly portrayed. Reviewer #2 is indeed asking to explicitly mention the source of the controversy regarding the interpretation of symmetry breaking in terms of charge ordering, mentioning that Ref. 26 reports evidence of oxygen order in partially reduced samples, rather than CDW. Moreover, from the additional comments of both reviewers #2 and #3, it appears likely that a similar scenario is found here, so this scenario should be clearly mentioned when discussing your results.

We are still very much interested in publishing your study in Communications Materials, but we would like you to take into account these considerations, revising the manuscript one last time following the latest recommendations, before we make a final decision on publication.

When submitting your revised manuscript, please include the following:

-A response letter with a point-by-point reply to each of the referee comments and a description of changes made. Please include the complete referee report in the response letter. Please note that the response letter must be separate to the cover letter to the editors.

-A marked-up version of the manuscript with all changes to the text in a different colored font. Please do not include tracked changes or comments. Please select the file type 'Revised Manuscript - Marked Up' when uploading the manuscript file to our online system.

-A clean version of the manuscript. Please select the file type 'Article File'.

-An updated [Editorial Policy](https://www.nature.com/documents/nr-editorial-policy-checklist.zip) checklist, uploaded as a 'Related Manuscript File' type. This checklist is to ensure your paper complies with all relevant editorial policies. If needed, please revise your manuscript in response to these points. Please note that this form is a dynamic 'smart pdf' and must therefore be downloaded and completed in Adobe Reader. Clicking this link will download a zip file containing the pdf.

In the event that your manuscript is accepted we will provide detailed guidance on our journal policies and formatting. You may however wish to ensure that the manuscript complies with our house style at this stage. See our style and formatting guide (<https://www.nature.com/documents/commsj-phys-style-formatting-guide-accept.pdf>) and checklist (<https://www.nature.com/documents/commsj-phys-style-formatting-checklist-article.pdf>) for reference.

Data availability statements and data citations policy: All Communications Materials manuscripts must include a section titled "Data Availability" at the end of the Methods section or main text (if no Methods). More information on this policy, and a list of examples, is available at <http://www.nature.com/authors/policies/data/data-availability-statements-data-citations.pdf>.

- Accession codes for deposited data
- Other unique identifiers (such as DOIs and hyperlinks for any other datasets)
- At a minimum, a statement confirming that all relevant data are available from the authors
- If applicable, a statement regarding data available with restrictions
- If a dataset has a Digital Object Identifier (DOI) as its unique identifier, we strongly encourage including this in the Reference list and citing the dataset in the Data Availability Statement.

DATA SOURCES: We strongly encourage authors to deposit all new data associated with the paper in a persistent repository where they can be freely and enduringly accessed. We recommend submitting the data to discipline-specific, community-recognized repositories, where possible and a list of recommended repositories is provided at <http://www.nature.com/sdata/policies/repositories>.

If a community resource is unavailable, data can be submitted to generalist repositories such as [figshare](https://figshare.com/) or [Dryad Digital Repository](http://datadryad.org/). Please provide a unique identifier for the data (for example a DOI or a permanent URL) in the data availability statement, if possible. If the repository does not provide identifiers, we encourage authors to supply the search terms that will return the data. For data that have been obtained from publically available sources, please provide a URL and the specific data product name in the data availability statement. Data with a DOI should be further cited in the methods reference section.

Please use the following link to submit your documents:

Link Redacted

We hope to receive your revised paper within three months; please let us know if you aren't able to submit it within this time so that we can discuss how best to proceed. If we don't hear from you, and the revision process takes significantly longer, we will close your file. In this event, we will still be happy to reconsider your paper at a later date, as long as nothing similar has been accepted for publication at Communications Materials or published elsewhere in the meantime.

Please do not hesitate to contact me if you have any questions or would like to discuss these revisions further. We look forward to seeing the revised manuscript and thank you for the opportunity to review your work.

Best regards,
Aldo

Dr Aldo Isidori
Senior Editor
Communications Materials

Reviewers' comments:

Reviewer #1 (Remarks to the Author):

In the first round of review, I raised a few critical questions based on my understanding. The authors have addressed these questions well. I also see adequate change of the manuscript in reply to my comments and questions. After the revision, the work is mainly focusing on the structural study of PNO and the explanation to the data is more concrete. Therefore I now recommend the work to be published on Communications Materials as it is.

Reviewer #2 (Remarks to the Author):

The authors have made substantive changes to the manuscript to focus on their main findings, removing key issues with the initial manuscript including speculation about this system being a potential route to investigating flat-band physics.

Despite these changes, the manuscript retains some deficiencies that have not been adequately addressed.

The introduction remains somewhat misleading regarding the interpretation of reports of charge order and oxygen order in the nickelates. I am in agreement with reviewer 3 that ref. 26 has provided compelling evidence that the $(1/3, 0, 1/3)$ peaks that were reported in ref.'s 15, 22–25 and are due to oxygen order in partially reduced samples, rather than an instability to CDW order, such as that found in the cuprates. It thus follows that a likely explanation of the new superstructure identified in this paper is also due to tendency for oxygen order in these materials. However the introduction continues to portray this key finding using vague language that “However, the interpretation of these results in terms of charge order is surrounded by controversy [26]”, as opposed to directly stating what the source of the controversy is. Note, the authors later argue that: “A single apical oxygen atom per $6 \times 6 \times 4$ (original) unit cells generates the observed symmetry breaking.” This is a very similar argument to that made in ref. 26 for the $(1/3, 0, 1/3)$ peaks, justifying a more direct and clear discussion of the findings from ref. 26.

Reviewer #3 (Remarks to the Author):

In my previous report, my main criticism was that the manuscript "has severe shortcomings in the context used to present the work and the arguments deployed for its importance." Although some helpful changes have been made, these edits are quite minor. Much more serious changes are required to make the paper suitable for publication, particularly in a moderately impactful journal like communications materials.

1. There is not adequate evidence that these measurements are important for PrNiO₂ itself. Indeed, the putative explanation is that the new superlattice phases are properties of PrNiO_x where x is substantially different from 2. The abstract, introduction, and (to some extent) the discussion implies that the work presents a study of "PrNiO₂". The manuscript does not adequately convey that this is likely a study of extrinsic phases arising from poorly controlled oxygen content in the synthesis of this material (I appreciate that the synthesis is difficult). The manuscript requires extensive editing to make it clear that the manuscript is likely studying extrinsic phases and not PrNiO₂ itself.

2. A reasonable work on the current topic must provide an introduction that weighs all of the literature on its current scientific merit. I still think the current introduction misleadingly implies that the manuscript will deliver interesting insights into correlated electronic physics. It talks a lot about analogies to cuprates. The manuscript does not provide any evidence that the actual observations reported provide meaningful insights to cuprates. The intro then talks about about electronic Hamiltonians and electronic symmetry breaking. The manuscript does not provide any evidence that the actual observations reported provide meaningful insights to electronic symmetry breaking --- and it explicitly suggests the opposite.

In my opinion, the papers suggesting that the $(1/3, 0)$ peaks in RENiO_x materials come from excess oxygen order are night and day more convincing than papers suggesting that this is electronic order (or hydrogen chains). Speaking bluntly, I think the authors are providing a very poorly balanced summary of the current research in a way that inappropriately implies that the manuscript will contain insights into electronic symmetry breaking.

3. I strongly believe that the appropriate context for this work, especially after the publication of Nature Materials, 1–6 (2024), is the tendency for perovskite-derived phases to form oxygen or oxygen vacancy order. I think these results on PrNiO_x should be made applicable to all perovskites, as the observed effects are far more likely to relate to atomic radii than anything that is intrinsic to the electronic properties of PrNiO₂. There is a substantial literature on this topic. You can refer to a review article here, for example: Chem. Mater. 1993, 5, 151-165. By saying this, I'm suggesting rewriting the introduction with oxygen vacancy order and perovskite as the primary topic --- this is the actual substance of this work. (Not just throwing in a citation.)

Communications Materials is committed to improving transparency in authorship. As part of our efforts in this direction, we are now requesting that all authors identified as 'corresponding author' create and link their Open Researcher and Contributor Identifier (ORCID) with their account on the Manuscript Tracking System prior to acceptance. ORCID helps the scientific community achieve unambiguous attribution of all scholarly contributions. You can create and link your ORCID from the home page of the Manuscript Tracking System by clicking on 'Modify my Springer Nature account' and following the instructions in the link below. Please also inform all co-authors that they can add their ORCIDs to their accounts and that

they must do so prior to acceptance.

Version 1:

Decision Letter:

Dear Mr Oppliger,

Thank you for submitting your revised manuscript, "Discovery of Giant Unit-Cell Super-Structure in the Infinite-Layer Nickelate PrNiO_{2+x}". In light of the revisions made to the paper in reply to the referees' previous comments, I am delighted to say that we are happy, in principle, to publish a suitably revised version in Communications Materials.

We therefore invite you to edit your manuscript to comply with our journal policies and formatting style in order to maximise the accessibility and therefore the impact of your work.

EDITORIAL REQUESTS

* Your manuscript should comply with our policies and format requirements, detailed in our style and formatting guide (<https://www.nature.com/documents/commsj-phys-style-formatting-guide-accept.pdf>).

* Please edit your manuscript according to the editorial requests in the attached table, and outline revisions made in the right hand column. If you have any questions or concerns about any of our requests, please do not hesitate to contact me. It is important that each request be addressed in order to avoid delays in accepting your manuscript. Please upload the completed table with your manuscript files as a Related Manuscript file.

* The editorial requests table also includes a full list of the files that must be provided upon resubmission. Please upload your files according to this table.

* An updated editorial policy checklist that verifies compliance with all required editorial policies must be completed and uploaded with the revised manuscript. All points on the policy checklist must be addressed; if needed, please revise your manuscript in response to these points. Please note that this form is a dynamic 'smart pdf' and must therefore be downloaded and completed in Adobe Reader. Clicking this link will download a zip file containing the pdf.

OPEN ACCESS

Communications Materials is a fully open access journal. Articles are made freely accessible on publication. For further information about article processing charges, open access funding, and advice and support from Nature Research, please visit <https://www.nature.com/commsmat/open-access>

Please use the following link to submit your revised files:

Link Redacted

We hope to hear from you within two weeks; please let us know if the process may take longer.

Best regards,

Dr Aldo Isidori
Senior Editor
Communications Materials

Authors replies to reviewers' comments:

Reviewer #1 (Remarks to the Author):

In the first round of review, I raised a few critical questions based on my understanding. The authors have addressed these questions well. I also see adequate change of the manuscript in reply to my comments and questions. After the revision, the work is mainly focusing on the structural study of PNO and the explanation to the data is more concrete. Therefore I now recommend the work to be published on Communications Materials as it is.

Authors: We thank the referee for the recommendation to publish our manuscript.

Reviewer #2 (Remarks to the Author):

The authors have made substantive changes to the manuscript to focus on their main findings, removing key issues with the initial manuscript including speculation about this system being a potential route to investigating flat-band physics.

Authors: We thank the referee for this second read and suggestions.

Despite these changes, the manuscript retains some deficiencies that have not been adequately addressed.

The introduction remains somewhat misleading regarding the interpretation of reports of charge order and oxygen order in the nickelates. I am in agreement with reviewer 3 that ref. 26 has provided compelling evidence that the $(1/3\ 0\ 1/3)$ peaks that were reported in ref.'s 15, 22–25 and are due to oxygen order in partially reduced samples, rather than an instability to CDW order, such as that found in the cuprates. It thus follows that a likely explanation of the new superstructure identified in this paper is also due to tendency for oxygen order in these materials. However the introduction continues to portray this key finding using vague language that “However, the interpretation of these results in terms of charge order is surrounded by controversy [26]”, as opposed to directly stating what the source of the controversy is. Note, the authors later argue that: “A single apical oxygen atom per $6\times 6\times 4$ (original) unit cells generates the observed symmetry breaking.” This is a very similar argument to that made in ref. 26 for the $(1/3\ 0\ 1/3)$ peaks, justifying a more direct and clear discussion of the findings from ref. 26.

Authors: We thank the referee for this comment. To address these points, we have now revised the introduction to place ref. 26 better in respect to previous publications. Additionally, we have extended the discussion to summarize previous x-ray scattering works (including ref. 26), their observations and interpretations. Here, we also emphasize our observations.

Reviewer #3 (Remarks to the Author):

In my previous report, my main criticism was that the manuscript "has severe shortcomings in the context used to present the work and the arguments deployed for its importance." Although some helpful changes have been made, these edits are quite minor. Much more serious changes are required to make the paper suitable for publication, particularly in a moderately impactful journal like communications materials.

1. There is not adequate evidence that these measurements are important for PrNiO₂ itself. Indeed, the putative explanation is that the new superlattice phases are properties of PrNiO_x where x is substantially different from 2. The abstract, introduction, and (to some extent) the discussion implies that the work presents a study of "PrNiO₂". The manuscript does not adequately convey that this is likely a study of extrinsic phases arising from poorly controlled oxygen content in the synthesis of this material (I appreciate that the synthesis is difficult). The manuscript requires extensive editing to make it clear that the manuscript is likely studying extrinsic phases and not PrNiO₂ itself.

Authors: We agree with the referee that the observed modulations are likely linked to oxygen diffusion and the observed broken translational symmetries indicate new PrNiO_{2+x} oxygen stoichiometries. To reflect this, we have adjusted the title and introduced abbreviations to emphasize that we refer to PrNiO_{2+x} instead of PrNiO₂.

We have also revised the manuscript to be more thoroughly labeled and described the three film systems used. Two of the films start in the PrNiO₂ structure but during the temperature annealing a new oxygen stoichiometry is likely reached. We thank the referee for helping us to improve the manuscript on this point.

2. A reasonable work on the current topic must provide an introduction that weighs all of the literature on its current scientific merit. I still think the current introduction misleadingly implies that the manuscript will deliver interesting insights into correlated electronic physics. It talks a lot about analogies to cuprates. The manuscript does not provide any evidence that the actual observations reported provide meaningful insights to cuprates. The intro then talks about about electronic Hamiltonians and electronic symmetry breaking. The manuscript does not provide any evidence that the actual observations reported provide meaningful insights to electronic symmetry breaking --- and it explicitly suggests the opposite.

In my opinion, the papers suggesting that the (1/3, 0) peaks in RENiO_x materials come from excess oxygen order are night and day more convincing than papers suggesting that this is electronic order (or hydrogen chains). Speaking bluntly, I think the authors are providing a very poorly balanced summary of the current research in a way that inappropriately implies that the manuscript will contain insights into electronic symmetry breaking.

Authors: In spirit of the referee comment, we have shortened the discussion of (electronic properties) cuprates. Furthermore, we now omit the mentioning of the Hubbard model.

3. I strongly believe that the appropriate context for this work, especially after the publication of Nature Materials, 1–6 (2024), is the tendency for perovskite-derived phases to form oxygen or oxygen vacancy order. I think these results on PrNiOx should be made applicable to all perovskites, as the observed effects are far more likely to relate to atomic radii than anything that is intrinsic to the electronic properties of PrNiO2. There is a substantial literature on this topic. You can refer to a review article here, for example: Chem. Mater. 1993, 5, 151-165. By saying this, I'm suggesting rewriting the introduction with oxygen vacancy order and perovskite as the primary topic --- this is the actual substance of this work. (Not just throwing in a citation.)

Authors: As mentioned under point 2, we have shortened the discussion of the cuprates and mentioning of the Hubbard model. In addition, we now explicitly explain, in the introduction, how the field moved from electronic to oxygen vacancy interpretations. Furthermore, we added a brief discussion of oxygen-ordering motifs to the introduction. We again thank the referee for the constructive comments that helped us to further improve the manuscript.